



# Groundwater-Surface water exchanges in an alluvial plain subjected to pumping: a coupled multitracer and modeling approach

Jérôme Texier[1], Julio Gonçalvès[1], Thomas Stieglitz[1], Christine Vallet-Coulomb[1], Jérôme Labille[1], Vincent Marc[2], Angélique Poulain[2], Philippe Dussouillez[2]

[1] Aix Marseille Univ, CNRS, IRD, INRA, Coll France, CEREGE, Aix-en-Provence, France
[2] UMR EMMAH Environnement Méditerranéen et Modélisation des Agro-Hydrosystèmes, University of Avignon, 84000 Avignon, France

*Correspondence to*: Jérôme Texier (texier@cerege.fr)

**Abstract.** Alluvial aquifers represent a vital water resource for many regions. However, understanding and characterizing
the interactions between rivers and these aquifers is a major challenge for researchers and water managers. This characterization, in terms of flow velocity or water supply, is important to identify the vulnerability of the aquifers. In this study, our goal is to improve the understanding of interactions between rivers and alluvial aquifers by combining a multi-tracer approach with numerical modeling. By integrating these two complementary methods, we aim to accurately quantify the exchanges between groundwater and surface water, and to identify the water sources contributing to aquifer recharge.
This combined approach allows a better quantification of river-aquifer interactions at local scale, in the context of groundwater exploitation by pumping along the river. A large drinking water catchment field located on the banks of the Rhône River, in the southeast of France, was chosen as the study site. This site consists of several pumping wells and observation piezometers parallel to the Rhône. As often, with alluvial aquifer exploitation close to a river, the pumping leads to flow from surface water to groundwater.
Groundwater temperature, piezometric levels and river surface water levels were continuously recorded for an 18-month period. During field campaigns, conductivity, stable isotopes of water and radon activity of groundwater and surface water were measured. Radon was applied in a new way to measure the water flow from the river to the aquifer, which reduces the natural radon signal in the aquifer by radon-poor waters from a river. On the site, the radon data clearly delineates groundwater recharge from the river within 50 meters from the banks. The methodology to interpret periodic groundwater
temperature signals was extended to the isotopic signal, making it possible to identify the dispersivity in addition to Darcy's velocity.
All the experimental data were accounted for in a synthetic MODFLOW model, taking into account the boundary effect of the Ouvèze and Rhone rivers. Model calibration was made using the piezometric records and the PEST package. Reactive transport of radon was implemented using MT3DMS to ascertain the overall water balance of the study site. In addition to
the Rhône as a supply, we have shown that the other river (Ouvèze) also contributes of the site's supply (around 55%). By improving our understanding of the interactions between rivers and alluvial aquifers, this study offers valuable insights for sustainable water resource management in regions similar to our study site, i.e., the case of a losing river recharging an exploited aquifer and demonstrates the value of natural tracers, such as radon or stable water isotopes, in situations where the





application of artificial tracers is impractical. Our results can guide policy decisions regarding groundwater development and
river ecosystem protection.

## 1 Introduction

When surface water and groundwater are connected, the abstraction of one of them can impact both resources and their
interactions. Modifying the direction and/or rates of their respective exchange fluxes can influence the long-term quantity
and quality of surface water and groundwater (Shu and Chen, 2002; Lu et al., 2018; Winter et al., 1999; Winter, 1999;
Sophocleous, 2002), affecting ecosystem equilibrium (Woessner, 2000), and downstream users (Winter et al. 1998). Alluvial
aquifers hydraulically connected to rivers are widely exploited and represent a major water resource. Although the proximity
to the river can ensure significantly high aquifer pumping rates, water quality problems can be encountered for those
pumping fields along the riverbanks (Bertln& Bourg, 1994; Hiscock & Grischek, 2002). Quantifying aquifer-surface water
bodies exchange is essential for understanding and joint management of surface water and groundwater resources
(Fleckenstein et al., 2010; Shanafield and Cook, 2014), as well as for possible contamination management (Chapman et al.,
2007; Chen, 2007; Lamontagne et al., 2005; Boano et al., 2010; Trauth et al., 2018).

The characterization of groundwater-river exchanges as well as methods to describe them, both in quantitative and
qualitative terms has been widely studied (see e.g., Bernard-Jannin et al., 2017; Brunner et al., 2009; Cardenas, 2009; Lu et
al., 2018; Rivière et al., 2014). Quantification of exchange is generally based on differential gauging or Darcy flow estimates
using piezometric records and hydraulic conductivity values (Dujardin et al., 2014; Keery et al., 2007; Schmidt et al., 2007;
Xie et al., 2016). However, these two approaches present serious limitations. Estimates of Darcy flows based on piezometric
level data are difficult due to problems in obtaining reliable hydraulic heads and hydraulic conductivity from piezometers,
especially in the case of a riverbed (Keery et al., 2007; Surridge et al., 2005). In addition, the hydraulic conductivity of
riverbed sediments is likely to vary by several orders of magnitude (Calver, 2001), adding further uncertainty to the Darcy
flux estimates. One of the most common methods for estimating river-aquifer exchange flow at the watershed scale is
differential gauging. This method is generally applied to rivers with low to moderate discharge (from 0.001 $m^3 s^{-1}$ to 500 $m^3$
$s^{-1}$; (Grapes et al., 2005; Konrad, 2006; McCallum et al., 2012; Opsahl et al., 2007; Xie et al., 2016). In the case of high flow
rivers, the use of this method is more difficult and must rely on fixed gauging stations rather than manual flow measurements
(Konrad, 2006; Xie et al., 2016).

Artificial tracers cannot always be used due to large river discharge or to the proximity to sensitive sites (industrial, urban),
so the use of natural tracer has been widely developed. Thermal signal transfer is a relevant tool for assessing water transfers
from rivers to unconfined aquifers that has attracted great attention over the past two decades (Constantz, 2008; Keery et al.,
2007). The advantage of temperature as a tracer, compared to hydraulic approaches, is that thermal properties vary less than
hydraulic properties, such as hydraulic conductivity, thus reducing the uncertainty of exchange flux estimates (Boano et al.,
2010; Lautz, 2010). The interest of temperature monitoring relies on its ease of implementation with inexpensive data





loggers, even over relatively long times and with fine time resolution (Constantz, 2008; Constantz and Stonestrom, 2003; Keery et al., 2007). Most studies focus on vertical temperature transfer from the river to the aquifer at the riverbed (Anderson, 2005; Constantz, 2008; Rosenberry et al., 2016; Ghysels et al., 2021; Xie and Batlle-Aguilar, 2017). Exchange fluxes are then estimated, by fitting an analytical solution to vertical riverbed temperature profiles (Schmidt et al., 2007;

Constantz, 2008), by analyzing time series of temperature data (Anibas et al., 2011; Rosenberry et al., 2016), or by means of coupled modeling of groundwater flow and heat transport (Anibas et al., 2011; Schmidt et al., 2007). Temperature is rarely used directly to quantify lateral (horizontal) exchange between streams and aquifers, but can be used as a calibration variable to constrain numerical models (Bravo et al., 2002; Brookfield and Sudicky, 2013; Tang et al., 2017). In these cases, piezometers and temperature sensors are often installed near the river (Engelhardt et al., 2011; Gerecht et al., 2011; Musial et

al., 2016).

Along with temperature, the use of radon is considered one of the most effective and well-established tracing techniques in hydrogeological studies (Kurylyk et al., 2018; Rau et al., 2010; Ren et al., 2018). This natural radioactive tracer is generally used to quantify the groundwater flow towards surface bodies such as coastal lagoons or rivers (Cook et al., 2006; Dugan et al., 2012; Hoehn and Von Gunten, 1989; Stellato et al., 2008; Cook et al., 2018; Stieglitz et al., 2013, 2010). Radon-222 is a

naturally occurring gaseous radioactive isotope with a short half-life (3.8 days). In groundwater, the activity of $^{222}$Rn is high, because it is produced by the decay of radium present in the soil minerals (Kraemer et al., 1998). Surface waters generally have low $^{222}$Rn activities due to the negligible radon content in precipitation, out gassing in the water column, and its short radioactive half-life (Cook et al., 2006; Cecil and Green, 2000; Bertln and Bourg, 1994; Dugan et al., 2012). This difference allows for radon to be used as a tracer of groundwater discharge to surface water bodies (Mayer et al., 2018; Stellato et al.,

2008; Hoehn and Von Gunten, 1989). Radon mass balance calculations enable the quantification of $^{222}$Rn and therefore groundwater fluxes upon quantification of radon concentration in groundwater (Cable et al., 1996; Cook et al., 2006; Corbett et al., 1997; Ellins et al., 1990). However, the situation of a losing river, i.e., surface water supplying the groundwater is less studied. In this situation, it is possible to obtain qualitative information on surface water groundwater exchange, when the boreholes are close to the river, and quantitative information, such as velocity and transit time, by analyzing the continuous

radon record (Hoehn and Von Gunten, 1989; Close et al., 2014).

Stable isotopes of water ($^{18}$O and $^2$H) are commonly used as conservative tracers in hydrology. Isotopic methods rely primarily on the Meteoric Water Line (MWL), which expresses a linear relationship between δ $^2$H and δ $^{18}$O values due to isotopic fractionation during condensation of water vapor mass (Craig, 1961), and represents the natural variations of the meteoric water composition before the influence of post-precipitation evaporation processes. The isotopic composition of

groundwater has been widely used to determine the origins of groundwater recharge, estimate mixing proportions between different sources, quantify recharge rates (Sharma and Hughes, 1985; Clark and Fritz, 1997; Gat et al., 1969; Gonçalvès et al., 2015; Séraphin et al., 2016; Vallet-Coulomb et al., 2017; Engelhardt et al., 2011; Fette et al., 2005). The application of stable water isotopes also makes it possible to improve numerical modeling of aquifer-river interactions, in particular by distinguishing different sources of water or by evaluating the recharge (Long and Putnam, 2004; Perrin et al., 2003; Binet et





al., 2017). Especially when contrasted signatures can be identified between a river and its accompanying alluvial aquifer, as is the case for rivers fed at high altitude, the application of stable isotopes of water also makes it possible to trace the surface water – groundwater exchanges and to determine the transit times (Poulain et al., 2021b, a).

Combining spatial and temporal, field and laboratory data with physically based numerical models is an active area of research and helps identify key dynamics and improve understanding of processes (Ghysels et al., 2021; Jiang et al., 2019; Binet et al., 2017; Fleckenstein et al., 2010; Gilfedder et al., 2015). Regarding groundwater-surface water exchanges, regional studies most often use MODFLOW type models with rivers represented by Cauchy type or prescribed hydraulic head boundary conditions (Rushton, 2007; Morel-Seytoux et al., 2014; Cousquer et al., 2017; Morel-Seytoux et al., 2018; Di Ciacca et al., 2019). However, a good calibration does not guarantee the reliability of a model, indeed several problems can arise as the possibility of multiple parameterizations (recharge, steady state permeability) to obtain the same result, the so-called equifinality issue (Cousquer et al., 2017; Bravo et al., 2002; Cousquer et al., 2018). In this context, the acquisition of constraining data (e.g., tracers) to increase the reliability of the variables and parameters (permeability, recharge, dispersivity…) is fundamental to improve the model quality (Delottier et al., 2017; Cousquer et al., 2018; Gardner et al., 2011; Fleckenstein et al., 2010). Although tracers can be individually used, the information obtained by combining different techniques (tracers) can validate the results and overcome the limitations of a single method (Dujardin et al., 2014; Gilfedder et al., 2015; Xie et al., 2016; Stellato et al., 2008; Sadat-noori et al., 2021; Gardner et al., 2011).

In this study, we combined a multitracer approach and numerical modeling in order to quantify the river-aquifer interactions at the local scale, in the context of groundwater exploitation by riverside pumping. A catchment field located on the banks of the Rhône River, in southeastern France, was chosen as the study site. This site has the particularity of being located between two rivers (Rhône and Ouvèze), thus likely prescribing hydraulic head values at river stages on each side if the rivers are connected to the exploited aquifer. As the pumping wells are located only a few dozen meters from the Rhône, there is a risk of pollution by the river. Piezometers and pumping wells were equipped with pressure and temperature sensors. In addition to continuous piezometric monitoring, various methods of evaluating the water exchanges between the alluvial aquifer and the river were implemented to obtain robust information on the nature and direction of the flows. The transfer of the seasonal (periodic) temperature signal from the Rhône River to the aquifer was monitored to quantify the lateral water exchange as well as the thermal parameters of the environment. Several radon and electric conductivity measurement campaigns were carried out to determine the nature of the exchanges in different operating situations (pumping or no pumping). The temporal monitoring of stable isotopes of water allows us to identify the different sources of water supplying the pumping wells and to characterize the mixing. By transposing the thermal periodic signal interpretation method and thus its analytical solution to the mass transfer problem, the stable isotopes of water time series analysis could enable quantifying the dispersivity of the medium. These different methods lead to a comprehensive understanding of the hydrodynamic behavior of the aquifer in a robust and repeatable way in the very common context of groundwater exploitation in an alluvial plain. All these data were aggregated and used to constrain a local hydrogeological model using MODFLOW and highlighting the vulnerabilities of the site regarding possible contamination coming from the rivers. To our knowledge, we propose several new approaches to





improve our understanding of alluvial aquifers in the context of river-groundwater exchange. While the standard use of
Radon is to obtain estimates of groundwater inflow in rivers, Radon has been used here in a novel way to determine the flow
of water from the river to the aquifer characterized by the dilution of the natural radon signal of the aquifer by the radon-
depleted waters of the rivers. To our knowledge, the use of radon data in transport models is rarely discussed and represents
a further addition to improve and constrain numerical models. In addition, we propose an extension of the standard 1D
interpretation of the periodic temperature signal to the periodic signal of stable water isotopes. Interpreting a natural tracer
such as stable isotope of water, in the context where the use of artificial tracers is not feasible, theoretically allows assessing
the poorly known dispersivity of the alluvial aquifer.

## 2 Study site, material and method

### 2.1 Field Site

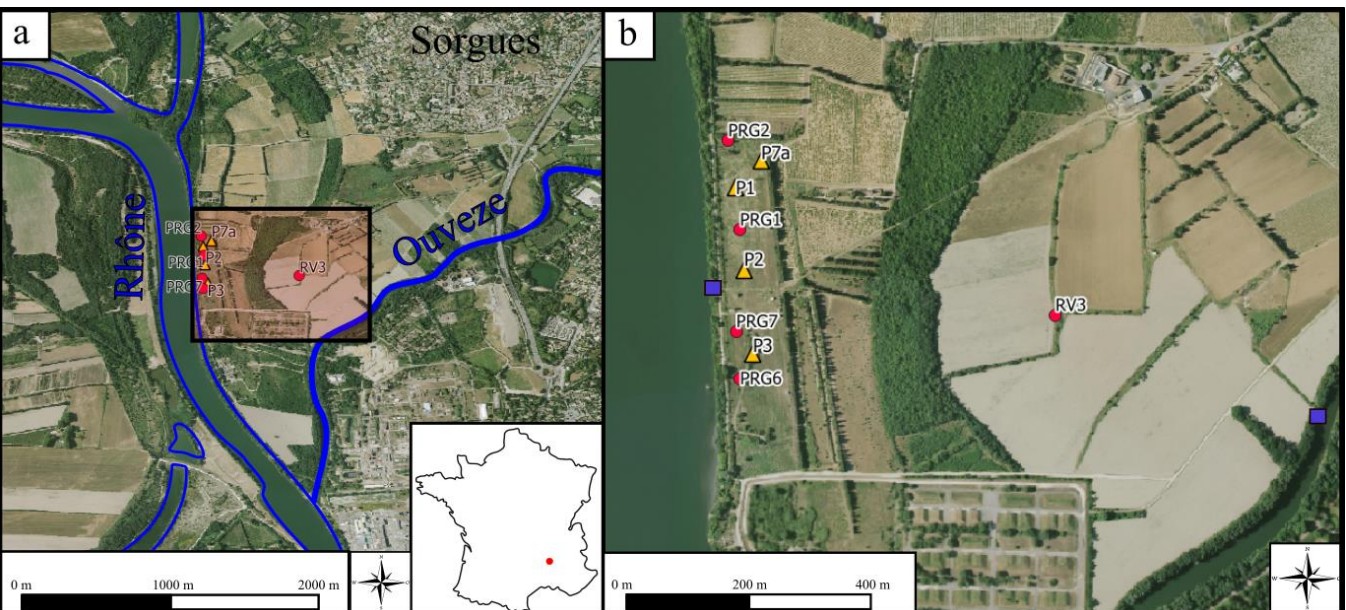

**Figure 1 : a) Map of the study area the Jouve site (Vaucluse, France) located in the red inset b) the monitored pumping wells and piezometers symbolized by yellow triangles and red points respectively. The water surface sampling sites are symbolized by blue squares (© Google Earth 2022).**

The study site located at Sorgues (Vaucluse, Southeast of France, Figure 1) is the domestic water extraction site of La Jouve
composed of pumping wells aligned along the Rhône riverbanks. The extraction site is located upstream of the confluence of
the Rhône and Ouvèze rivers, two particularly important surface water bodies, which results in a great local stability of
groundwater levels. The site is equipped with piezometers allowing the monitoring of the water table and of four active
pumping wells operating intermittently for a total extraction flow of 11,000 m$^3$ per day. The exploited unconfined alluvial
aquifer corresponds to a plio-quaternary formation mainly composed of sand and pebbles deposited by the Rhône River. The



aquifer substratum at the extraction site is composed of Miocene blue marls from between an elevation of 9 and 11m absl
(Caridroit, 1968; Nofal, 2014). For the Rhône, the level is controlled upstream at the northwest by the Sauveterre dam which
maintains the river stage at about 16.5 m absl.

## 2.2 Material

The site was equipped with a set of pressure and temperature sensors in February 2019. HOBO Water Temp Pro v2
temperature sensors were installed vertically in the pumping wells (P1, P7A, P3; see Figure 1), with the groundwater level
being provided by the pumping wells operator. The accuracy, resolution and response time of the probes are 0.2 °C, 0.02 °C
and 5 min, respectively. The site's observation piezometers are monitored for pressure and temperature by OTT Orpheus
Mini probes with an accuracy of 0.05% for water-level measurement and a resolution of 0.1 °C and an accuracy of 0.5 °C for
temperature.

Four piezometers on the pumping site have been equipped, as well as one outside (200 m to the East, RV3 see Figure 1). The
temperature and the river stage of the Rhône were also monitored. All probes have been configured to record measurements
every 15 minutes.

Radon was sampled in surface water and groundwater during three campaigns: March 2020, June 2020 and April 2021. The
first campaign focused on the Rhône riverbank and the pumping site during a period of pumping shutdown. The second
campaign took place when pumping activity was restored and focused on the groundwater abstraction site. The third
campaign focused on the Ouvèze riverbank and the aquifer outside the site in the same hydrodynamic context of the second
campaign (pumping wells functioning and related pseudo-steady state). The radon activity of the samples was measured at
the laboratory using electronic radon-in-air monitors (Durridge RAD-7). Radon dissolved in water is circulated through an
air loop and then through the monitors to establish an equilibrium between air and water (Stieglitz et al. 2010, for example).
The monitors count α decays of $^{222}Rn$ progeny, and $^{222}Rn$ activity is determined by discriminating $^{222}Rn$ progeny decays in
specific energy windows.

Surface water and groundwater were collected between June 2019 and November 2020 for water stable isotope analysis
($^{18}O$, $^{2}H$). Groundwater samples were collected in the pumping wells and piezometers by means of a submersible
groundwater pump and access taps. For the piezometer a pumping time equivalent to the displacement of 3 borehole
volumes resulted in a constant value of specific conductance. The composition in $\delta^{18}O$ and $\delta^{2}H$ of the water samples were
analyzed using a Picarro L2140-i wavelength-scanned cavity ring-down spectroscopy (CRDS), following the protocols
described in Vallet-Coulomb et al. 2021 for post-measurement data processing. All samples were filtered before analysis and
analyzed twice. All data are provided in ‰ *vs* VSMOW, after normalization to the VSMOW2-VSLAP scale. Based on the
performances obtained for samples of a laboratory standard included in each analysis runs as a quality assurance / quality
control, the precision obtained was 0.03‰ and 0.09‰ for $\delta^{18}O$ and $\delta^{2}H$ respectively (n = 6 runs). The Rhône isotopic
signature records were obtained by the Avignon databases (Poulain et al., 2021a, b).



### 2.3 Methodological approaches

#### 2.3.1 1D temperature analysis

Heat transfer in a half-space (river infinite porous medium) is assumed to be governed by a one-dimensional conduction-advection equation, and is usually solved in the framework vertical transfers from the streambed downwards (Stallman, 190  1965; Hatch et al., 2006) (1).

$$\frac{\partial T}{\partial t} = Ke\frac{\partial^2 T}{\partial z^2} - \frac{nv_f}{\gamma}\frac{\partial T}{\partial z}\ (1)$$

where $T$ is temperature (°C) variable with time $t$ (s), and depth z (m), $Ke$ is the effective thermal diffusivity, $\gamma = \frac{\rho c}{\rho_f c_f}$ the ratio of the heat capacity (-) of the streambed to that of the fluid, $n$ is the kinematic porosity (-), and $v_f$ is the vertical fluid velocity m.s⁻¹.

The heat transfer from the Rhône River to the aquifer was studied by adapting the one-dimensional (1D) solution to calculate the amplitude change of a periodic temperature signal (seasonal signal of the surface water) with the distance to the riverbed and the shift when this signal is propagating within the aquifer (Goto and Matsubayashi, 2009; Stallman, 1965). This analytical solution, in its standard form, assumes 1D uniform vertical flow, sinusoidal behavior of the surface temperature, and no change in the average temperature with depth of the response time to the temperature variation at the surface (Hatch 200  et al., 2006). Here, the solutions of Goto (2005) and Stallman (1965) have been adapted to study the attenuation and phase shift in the case of a purely horizontal (x-axis) transfer from a riverbank and writes (2):

$$T(x,t) = \text{Aexp}\left(\frac{vx}{2Ke} - \frac{x}{2Ke}\sqrt{\frac{\alpha+v^2}{2}}\right)cos\left(\frac{2\pi t}{P} - \frac{x}{2Ke}\sqrt{\frac{\alpha-v^2}{2}}\right)\ (2)$$

where $A$ is the amplitude of the temperature variations (-), $P$ is the period of temperature variations (s), $v$ is the horizontal velocity (m.s⁻¹) and $\alpha = \sqrt{v^4 + \left(\frac{8\pi Ke}{P}\right)^2}$

An inversion using the piezometer records allowed us to identify the hydraulic and thermal parameters (velocity and thermal diffusivity of the medium).

#### 2.3.2 1D Stable isotope transport

The lateral transfer of surface water into the alluvial aquifer can also be studied using the periodic signal of the stable isotopes of water. For this purpose, the sinusoidal fluctuations of the isotope concentration can be studied by adapting to 210  mass transport the previous solution for heat transfer. By parameter identification of the 1D dispersion advection equation (3) which writes:

$$\frac{\partial c}{\partial t} = D\frac{\partial^2 c}{\partial x^2} - v_f\frac{\partial T}{\partial x}(3)$$

and heat transfer equation (1), the analytical equation (2) was transposed to a mass transport problem giving (4):





$$C(x,t) = \text{Aexp}\left(\frac{\frac{v_f}{n}x}{2D} - \frac{x}{2D}\sqrt{\frac{\alpha + \frac{v_f^2}{n}}{2}}\right)cos\left(\frac{2\pi t}{P} - \frac{x}{2D}\sqrt{\frac{\alpha - \frac{v_f^2}{n}}{2}}\right)(4)$$

Eq. (4) was applied on the stable of water monitoring. Using the Darcy velocity obtained from temperature signal interpretation (eq. 2), eq. 4 allows identifying the dispersion coefficients (D).

**2.3.2 Coupled Flow and Radon Transport Model**

In order to integrate the results obtained with the different tracers and to estimate the vulnerability to contamination of the
site, a synthetic model was built using MODFLOW. The numerical codes used for groundwater flow and transport modeling are MODFLOW (Mcdonald and Harbaugh, 1988) for groundwater flow and MT3DMS (Zheng and Wang, 1999) for advective-dispersive transport integrated into the PROCESSING MODFLOW X software. MODFLOW solves the diffusivity equation using a finite difference method. Thus, for horizontal 2D flows in a free, heterogeneous and isotropic aquifer, the diffusivity equation writes:

$$\omega_s\frac{\partial h}{\partial t} + Q(x,y) = \frac{\partial}{x}\left(T(x,y)\frac{\partial h}{\partial x}\right) + \frac{\partial h}{\partial y}\left(T(x,y)\frac{\partial h}{\partial y}\right)(5)$$

With $\omega_s$ (-) the specific yeld, $h$ (m) the hydraulic head, $T(x, y)$ (m² s⁻¹) the aquifer transmissivity, $Q(x, y)$ (m s⁻¹) source or sink term.

MT3DMS was used for radon transport calculations. MT3DMS allows the simulation of dissolved species concentrations by considering advection, dispersion, molecular diffusion, and chemical reaction (Zheng and Wang, 1999; Zheng et al., 2012).
MT3DMS thus relies on the flow solutions of MODFLOW to solve the advection-dispersion equation:

$$\frac{\partial(\theta C)}{\partial t} = \frac{\partial}{\partial x_i}\left(\theta x_{ij}\frac{\theta C}{\partial x_j}\right) - \theta v_i\frac{\theta C}{\partial x_i} + q_s C_s + \sum R_n (6)$$

where $C$ (Kg m⁻³) the concentration of the dissolved species, $\theta$ (-) the aquifer porosity, $x_i$ (m) the distance belong an axe, $D_{ij}$ (m² s⁻¹) the dispersion coefficient, $v$ (m s⁻¹) the pore velocity, $q_s$ (s⁻¹) volumetric flow rate per unit volume of the source or sink, $\Sigma R_n$ (g m³ s⁻¹) the chemical reactions term.

The modeled surface area is 3 km² and covers the Jouve site and the two rivers, the Rhône and the Ouvèze. The boundary conditions of the model are prescribed hydraulic head conditions: the Rhône at the western limit set at 16.5 m, the Ouvèze at the east set at 18.2 m and a northern limit at 17 m, considered time invariant according to the piezometric monitoring of the pumping field operator. The model is composed of 2600 rows and 2200 columns with 5m wide grid cells. The model includes an aquifer layer limited by a lower surface corresponding to the alluvial substratum and an upper surface
corresponding to the topography. An annual recharge of 300 mm years by direct rainwater infiltration was introduced  based on previous estimates in the Avignon region (Nofal, 2014). The calibration to adjust the permeability field of the model to reproduce the measured piezometric levels was performed at  steady-state using the PEST optimization tool (Doherty, 2004). Initial permeability values were obtained by interpolation using data from the operator's hydraulic test reports (value of 1.10⁻





$^2$ to $1.10^{-4}$ m s$^{-1}$). Radon transport was integrated into the model by including the radioactive half-life of radon (3,823 days)
in MT3D and by setting the rivers at a constant concentration of radon measured (200 Bq m$^{-3}$). A dispersivity of 10 meters
was used based on usually found values in the literature for sandy alluvial aquifers at this scale (Schulze-Makuch, 2005).
The geological media generation of radon (source term) was incorporated into the model by using the flow injection wells
package for each cell of the aquifer which enabled reproducing the background radon signal in the aquifer. The injection rate
is very low to prevent any impact on the piezometric level. The radioactive decay of radon was added in the model and the
radon injection concentration (geological generation) is then optimized using PEST to reproduce the radon activity observed
between the Rhône and the pumping wells.

## 3 Results

### 3.1 Piezometric level variations

The piezometric level variations at the site (Figure 2) are controlled at short time scale by the pumping cycles of the four
wells (P1, P2, P3 and P7A). Over a period of continuous pumping, from January 2019 to February 2020, the pumping
maintains the water level between 15.5 m and 14.6 m, except during flooding events of the Rhône River. A pseudo-steady
state is therefore established at the site caused by periodic pumping. At longer time scale, river stage variations of the Rhône
influence the water table, particularly during periods of flooding marked by a sharp increase in the level of the Rhône
(Figure 2). When the site is operative, the water table is permanently below the Rhône River stage, even during short
pumping stops and floods. The conditions are therefore favorable for an exchange from the Rhône to the alluvial aquifer.
Site maintenance operation led to a prolonged pumping stop (6 weeks during spring 2020, red period, Figure 2). During this
period, the water table rose progressively until reaching a stable level above the Rhône River stage, and the exchanging flow
between the Rhône and the alluvial aquifer likely reversed. During periods of flooding, the Rhône River rises to an average
value of 16.9 m. Higher values were observed in October and December 2019 with sharp increases to 18.9 m and 19.4 m
respectively as a result of water releases from the Sauveterre dam (Figure 2). During the aquifer pumping period, the
piezometric levels pointed to flow from the Rhône and the Ouvèze towards the Jouve site (Figure 3).





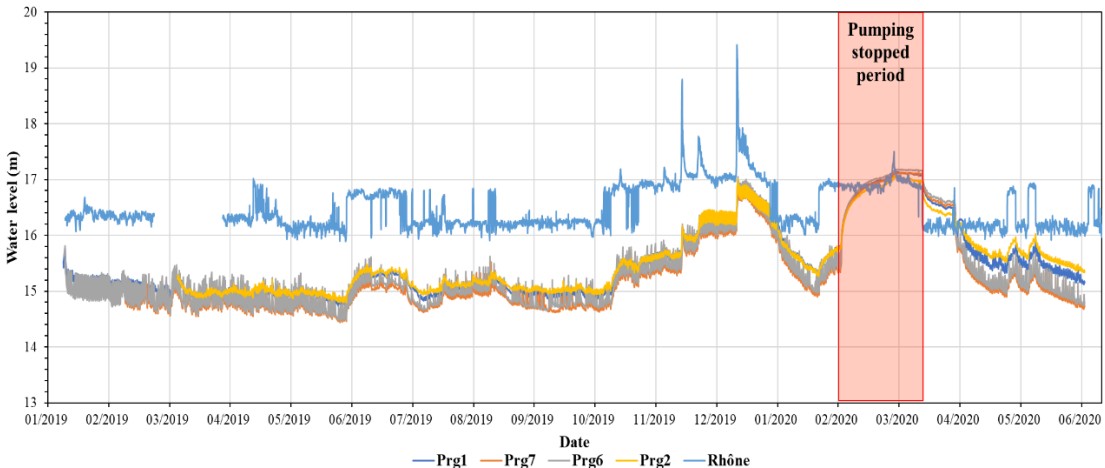

**Figure 2 : Piezometric level time series at the study site of la Jouve (Prg1, Prg7, Prg6, Prg2) and Rhône River stage time series at 15 min intervals.**

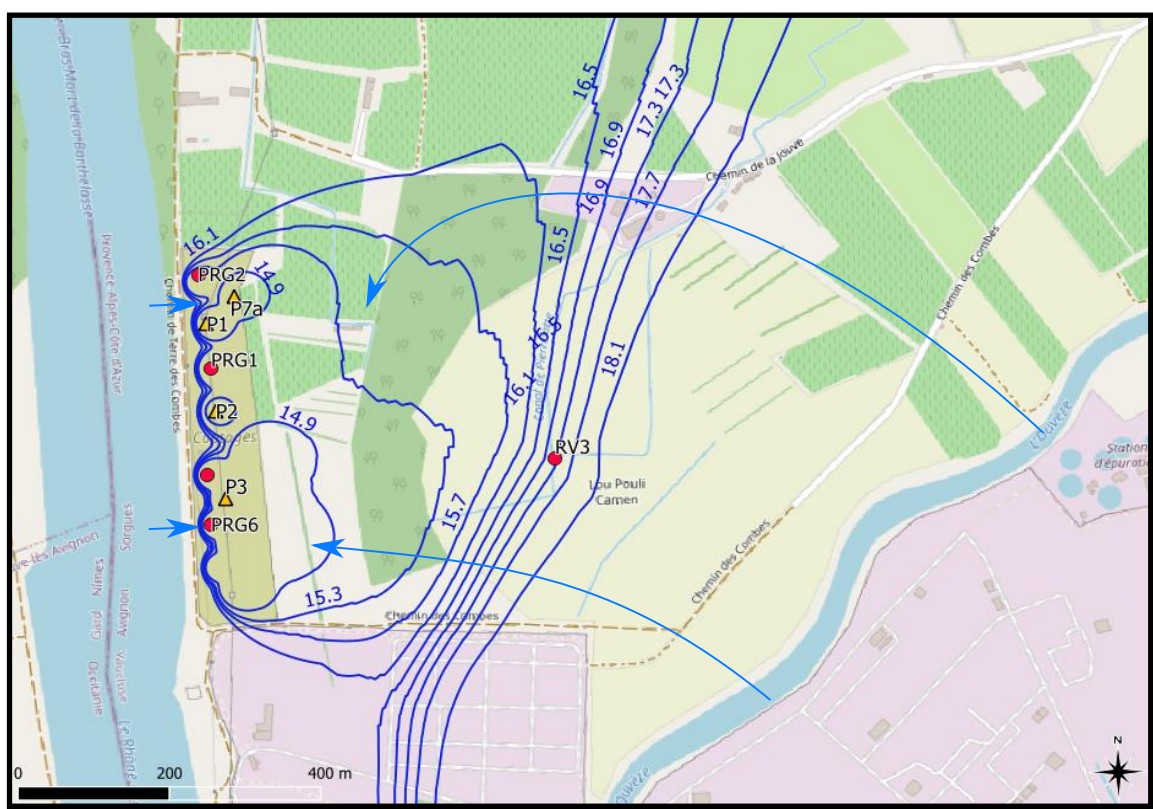


**Figure 3 : Piezometric map obtained by interpolation of piezometric level using the inverse distance to a power method (campaign during October 2018). © OpenStreetMap contributors 2022. Distributed under the Open Data Commons Open Database License (ODbL) v1.0.**

## 3.2 Interpretation of Radon and Electrical conductivity data

During the first field campaign, when the pumping wells were stopped, the radon signature of the Rhône was about 200 Bq m$^{-3}$ and at the Prg2 piezometer, the value was at 12,000 Bq m$^{-3}$ (Figure 4). During the second campaign, pumping was restored and the usual piezometric levels were recovered; the same piezometer, Prg2, showed Radon activities four times lower at 3,000 Bq m$^{-3}$. The other piezometers, close to the Rhône, showed similar low values, while the pumping wells further away were at values of between 10,000 Bq m$^{-3}$ and 8,000 Bq m$^{-3}$. These latter values were close to those found

during the first campaign at the SRV3 piezometer outside the site. During the third campaign, the hydrodynamic conditions on the site were the same as during the second campaign (pumping in operation). The piezometers near the Ouvèze had low radon values at about 3,000 Bq m$^{-3}$ compared to 11,000 Bq m$^{-3}$ at a downstream distance of 100 m away from the Ouvèze in the aquifer.

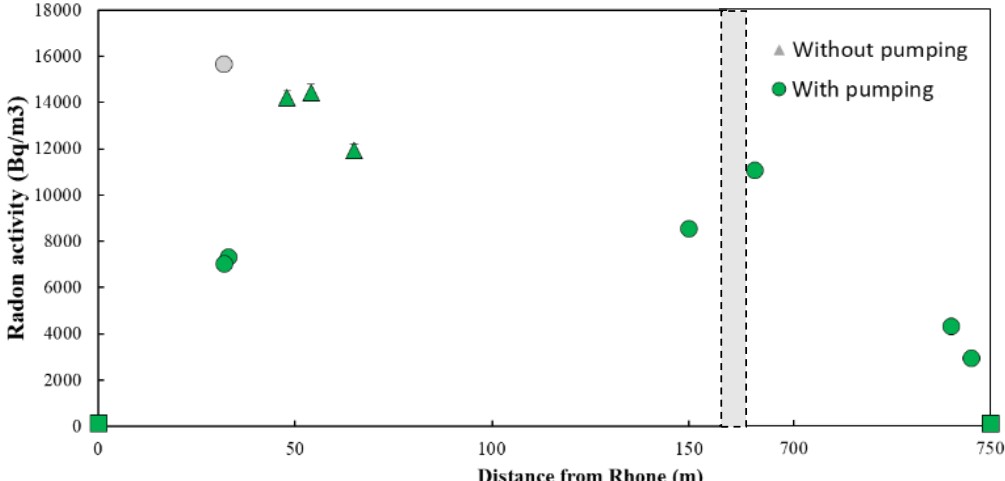

**Figure 4 : Radon activity as a function of the distance from the Rhône riverbank (Rhône at 0 m and Ouvèze at 750 m).**

In comparison, the electrical conductivity of water samples measured during the second and third campaigns showed a similar behavior as radon during pumping, with almost constant values in the aquifer at about 680 mS cm$^{-1}$, the Rhône at 350 mS cm$^{-1}$ and the Ouvèze at 570 mS cm$^{-1}$ (Figure 5). The wells and piezometers showed intermediate values between those observed in the aquifer and in the Rhône, with decreasing values for the wells and piezometers closer to the Rhône.





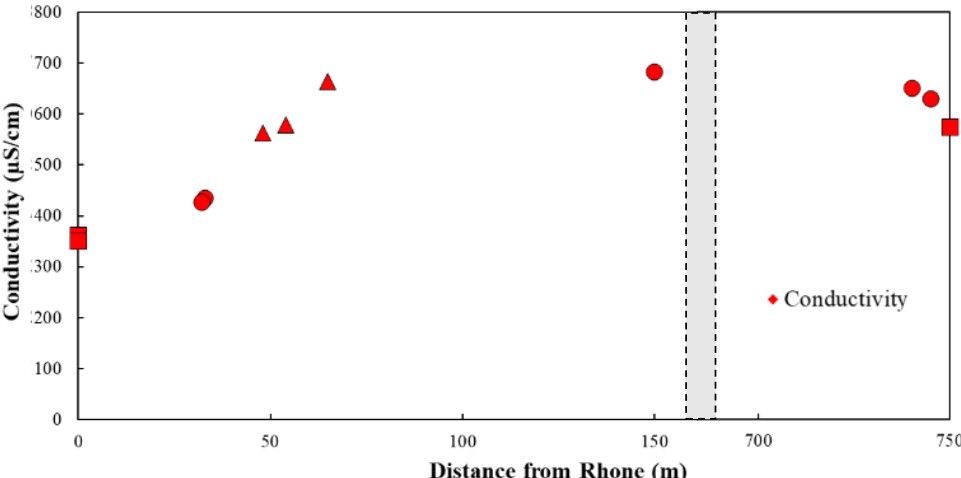


**Figure 5 : Electrical conductivity of groundwater and surface water as a function of the distance to the Rhône (Rhône at 0 m and Ouvèze River at 750 m).**

### 3.3 Stable isotopes of water

The two rivers delimiting the study area, the Rhône and the Ouvèze, present depleted isotopic compositions compared to local precipitation, attributed to the altitudes of their respective catchments. The groundwater isotopic compositions are spread around the GMWL, in the range of the composition of these two rivers, with a small influence of evaporation for some of the samples. As previously described, the sampling protocols were carefully implemented, and we are confident that the sample compositions were representative of the groundwater composition (sufficient purging phase for piezometers, and

a continuous flux for pumped wells). The influence of a direct evaporation of groundwater is possible, considering the shallow depth of the vadose zone (4-5m), and the particularly dry Mediterranean climate of the area (Coudrain-Ribstein et al., 1998).

Considering the composition of local precipitation, the potential influence of a local groundwater recharge remains undetectable. The piezometers located close to the Rhône (PRG2: 42 m, PRG6: 33 m and PRG7: 35 m) present isotopic

compositions fully compatible with a lateral recharge coming from the Rhône River, and further modified by the influence of evaporation. The spread of the data can be explained by the seasonal variation of the Rhône composition, and a variable contribution of evaporation.

The distances between the pumping wells and the Rhône River are slightly higher (P1 : 56 m, P2 : 58 m, P7a : 97 m), and their compositions are slightly enriched compared to that of these piezometers. This could be explained by a small

contribution of the Ouvèze River, in addition to that of the Rhône. This will be discussed latter.





On the other hand, the composition of the piezometer RV3, located in between the two rivers, points towards a dominant contribution of the Ouvèze River.

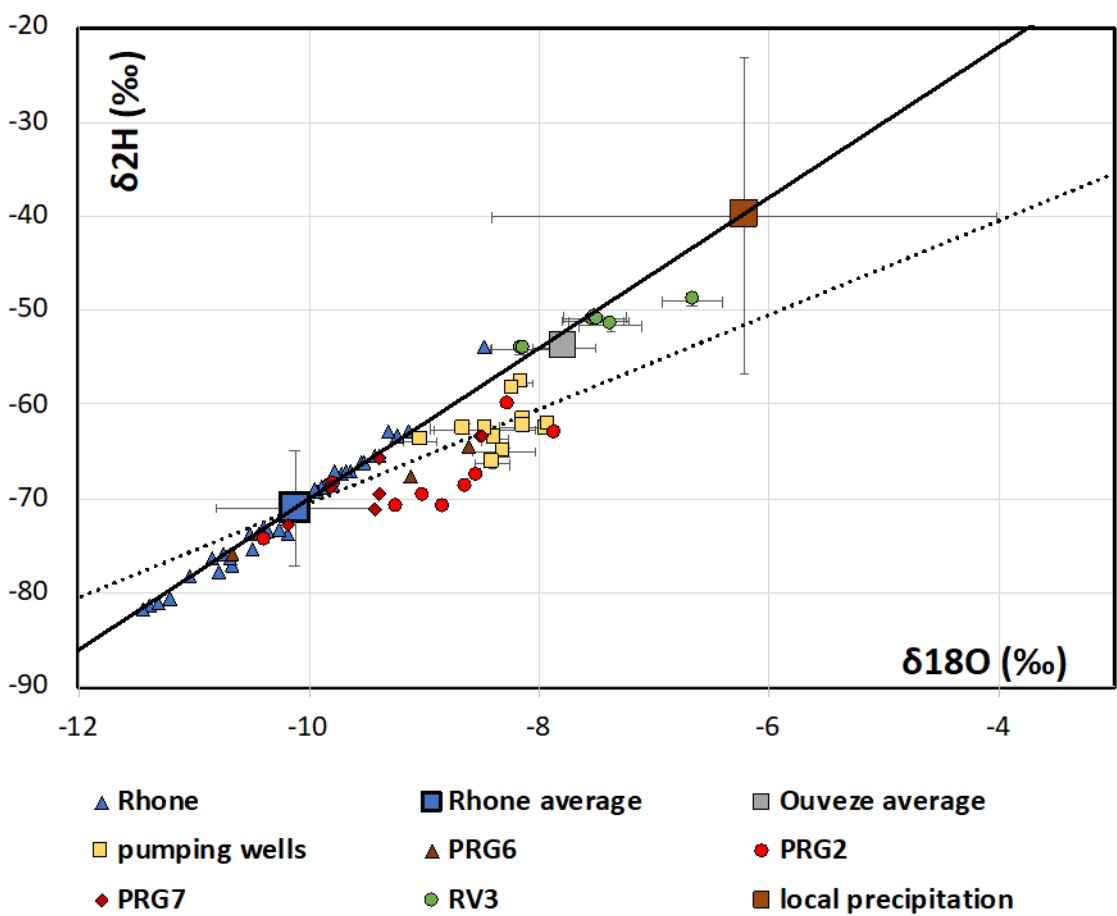

**Figure 6 : Isotopic compositions of local precipitation (long term weighted average of the Avignon GNIP station), surface water (Rhône and Ouvèze rivers) and groundwater sampled in piezometers and pumping wells (average values of each sampling campaign). The Rhône's isotopic data are obtained from Poulain et al. (2021a). Error bars represent the standard deviations of average values. The black line represents the global meteoric water line. The dotted line represents a theoretical evaporation line crossing the Rhône average composition, and assuming a slope of 5.**

### 3.4 Interpretation of Temperature and Isotopes with Periodic Signals

Temperature monitoring over the same period showed a periodic variation at all wells and piezometers as well as in the Rhône. The variations in the piezometers showed a significant attenuation as well as a phase shift with respect to the Rhône River temperature signal. During the period when the pumping wells were stopped, discontinuities appeared in the data due to the change in the hydraulic regime (Figure 7). The monitoring of temperatures, over the period not impacted by the maintenance works, allows the use of the 1D solutions described in Section 2.3.





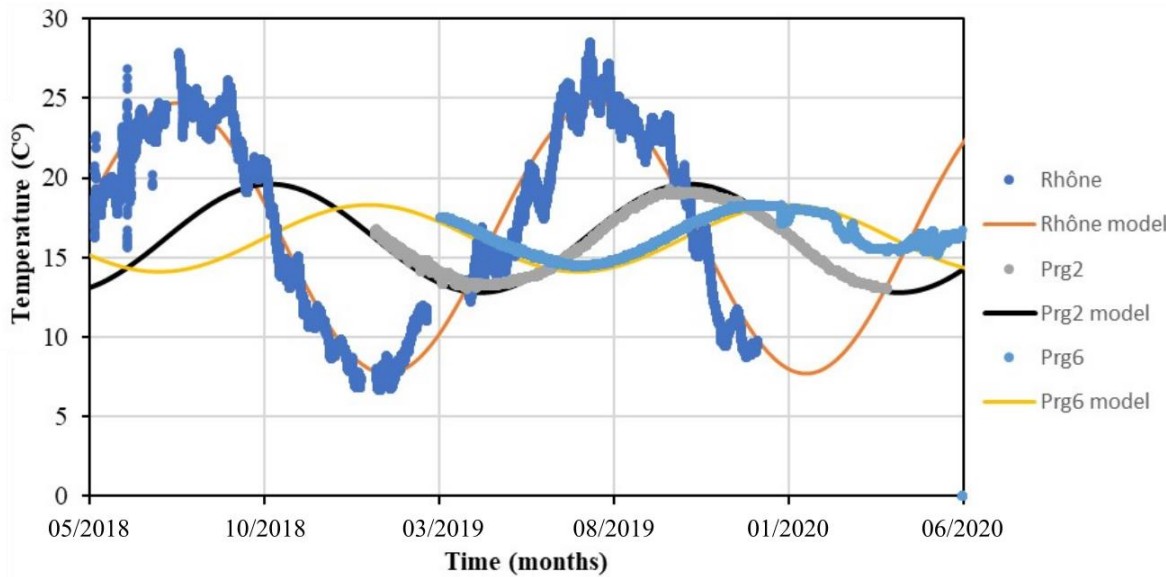


**Figure 7 : Temperature monitoring of piezometers Prg2 and Prg6 and the Rhône River and corresponding fitted 1D analytical expressions.**

Upon calibration, the 1D analytical expression for temperature provides the thermal front velocity and by extension the fluid flow velocities. The pore velocity values between $4.10^{-5}$ m s$^{-1}$ and $3.10^{-6}$ m s$^{-1}$ were obtained with a porosity of 10% (value

from the operator). The lower velocity was obtained for Prg1 while the three other piezometers led to velocities in the order of $10^{-5}$ m s$^{-1}$ (Table 1). These values are smaller than the first estimate made by calculating Darcy's flux based on the site permeability data (average value of $1.3\ 10^{-3}$ m s$^{-1}$) and the hydraulic head gradient and leading to values ranging from $1.10^{-4}$ to $3.10^{-4}$ m s$^{-1}$. This difference can be explained by the fact that the heat transport accounts for the entire continuum between the river and the aquifer with possible heterogeneity, while the Darcy method considers a homogeneous aquifer. Thus, the

lower velocities obtained by this method can be explained by the presence of finer materials along the banks or a partial clogging of the riverbed, reducing the hydraulic conductivity.

**Table 1 : Calculated pore fluid velocity and thermal parameters based on the 1D periodic temperature analytical solution.**

| Wells | Distance (m) | Pore velocity (m s$^{-1}$) | Ke(m s$^{-1}$) |
|-------|--------------|----------------------------|----------------|
| Prg2 | 42 | $4.2.10^{-05}$ | $5.8.10^{-05}$ |
| Prg7 | 35 | $1.1.10^{-05}$ | $2.3.10^{-05}$ |
| Prg6 | 33 | $3.10.10^{-05}$ | $1.5.10^{-05}$ |
| Prg1 | 45 | $3.45.10^{-06}$ | $1.9.10^{-05}$ |





The isotopic signal phase shift could also be interpreted using Equation 3 presented in the methodology section. However, due to the evaporative signature of the groundwater samples from the pumping site and to a relatively low sampling frequency, the reliability of the observed seasonal signal in the aquifer is more questionable, contrary to the Rhône River. An exploratory application using the proposed solution (Eq. 3) is nevertheless made in the complementary section (Appendix A) but remains almost theoretical due to data evaporation bias.

## 3.5 Fluid flow and transport modeling of the alluvial aquifer at the study site

### 3.5.1 Fluid Flow modeling and calibration

The hydraulic conductivity calibration consists in adjusting the permeability of the model, to reproduce the measured piezometric levels. The PEST package was used to calibrate the model together with the average piezometric value during the site activity (pseudo-steady state). The calibration allows reproducing the groundwater level (Figure 8) and provides
hydraulic conductivity values between $1.10^{-2}$ and $1.10^{-4}$ m s$^{-1}$ for the aquifer and lower value of $5.10^{-5}$ for the Rhone bank (values consistent with the operator reports).

Modeling under pumping conditions results in a calculated flow from the Ouvèze to the well field, as well as from the Rhône to the well field. In accordance with the piezometric monitoring, the piezometric depression is deeper in the southern part of the site during pumping. When pumping is active, the site is fed on the east side by the aquifer itself fed by the Ouvèze and
on the west side directly by the Rhône (Figure 8A). Without pumping, the natural piezometric level is consistent with the piezometric monitoring during the pumping stop period with a reversal of the Rhône-aquifer exchange at the site. In fact, in this configuration, the flow is uniformly directed from the Ouvèze to the Rhône, including at the extraction site (Figure 8B).



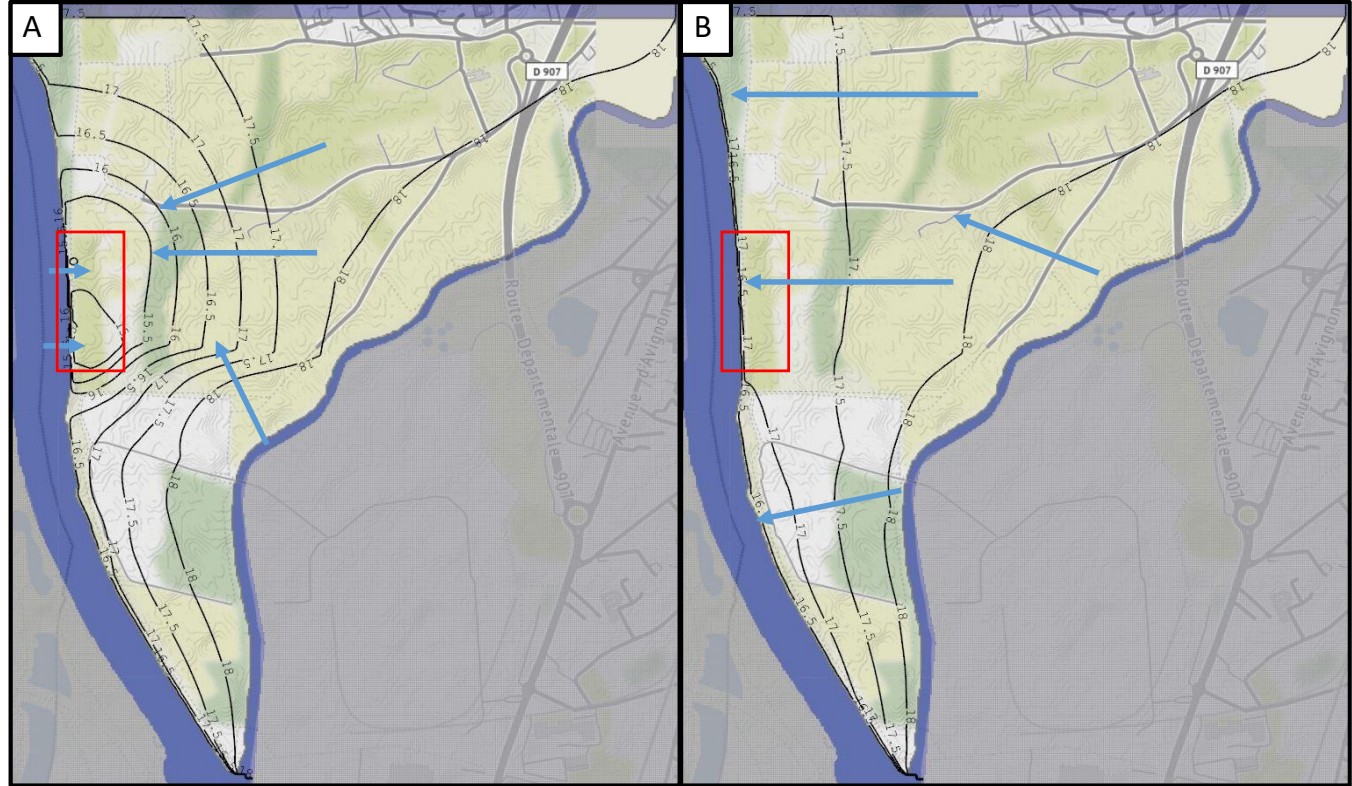

**Figure 8 : A) Steady-state piezometric map simulated with MODFLOW for the area around the Jouve site with active pumping**
**(period from 01/2019 to 02/2020) without flooding of the Rhône (RMSE=0.018 m); B) steady state piezometric map simulated**
**using MODFLOW without pumping and thus corresponding to the natural regime. The Jouve extraction site area is within the**
**red rectangle. C) Observed value and calculated value of the piezometric level in the Jouve site during pumping after hydraulic**
**conductivity calibration using PEST.**

### 3.5.2 Radon Transport Modelling

Introducing the reactive transport of the radon enables producing a profile of the aquifer from the Rhône to the well. Figure 9
shows that the MT3D transport model accurately reproduced the observations obtained during the different campaigns under
the same hydrodynamic conditions, with a gradual increase in radon activity in the aquifer during pumping, due to the
progressive increase in radon concentration in the water coming from the Rhône due to gas production within the aquifer.
The radon activity increases to a maximum of about 12,000 Bq m$^{-3}$ at about 60 m of the Rhone River.






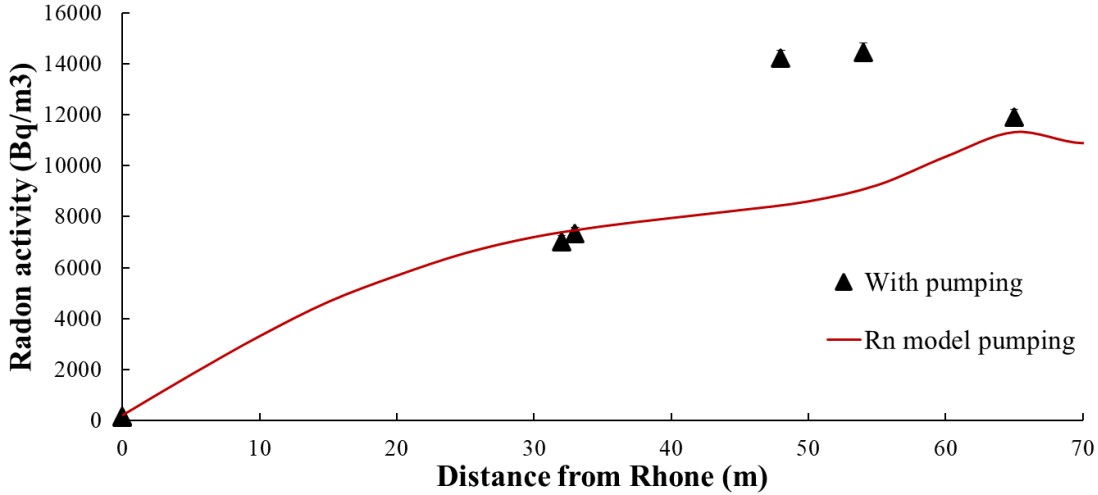

**Figure 9 : Comparison between measured and simulated radon activities in the Rhône River, piezometers and pumping wells in pumping conditions.**

## 4 Discussion

The groundwater is characterized by variable radon activities between 8,000 Bq m$^{-3}$ and 15,000 Bq m$^{-3}$ which can be explained by some heterogeneity in internal radon production due to petrographic variability. During the pumping period, the radon measurement indicated groundwater recharge from both rivers, i.e., the Rhône and the Ouvèze, thus inducing low radon activities during pumping. These supplies are confirmed by radon in the piezometers near the Rhône and the Ouvèze, even if due to the progressive enrichment of radon in the aquifer, the activity returns to an equilibrium value in the order of

12,000 Bq m$^{-3}$ from 50 m downstream. As the wells are located between 50 and 90 meters of Rhône, at this distance the dilution effect of radon activity is no longer visible in the wells, contrary to the piezometers. For the Ouvèze, as for the Rhône riverbank, the lower radon signal caused by the intrusion of the river water with a low radon concentration disappears moving away from the river. We can deduce that the pumping wells producing the piezometric depression have a multiple supply with one part coming from the Rhône River and another one from the groundwater supplied by the Ouvèze. This

hypothesis is supported by the piezometric contours that shows a flow from the Ouvèze towards the Rhône either in pumping or no pumping conditions. However, the radon activity does not allow for total quantification of recharge.

This double supply to the pumping wells is also well established by the δ$^{18}$ O-δ$^{2}$H data, indicating an intermediate composition between that of the Rhône and that of the Ouvèze. Indeed, while the piezometers located near the Rhône have isotopic compositions that are entirely compatible with lateral recharge from the Rhône and modified by the influence of

evaporation. The isotopic compositions of the pumping wells are slightly enriched compared to those of these piezometers. This is explained by a contribution from the Ouvèze, in addition to that of the Rhône. This is confirmed by the composition of well RV3, located between the two rivers on the flow line from the Ouvèze supplying the pumping site, which indicates a



dominant contribution from the Ouvèze. In this first approach, the global recharge was considered here as negligible with a value at 300 mm for an area of 3 km² and a total pumping of 12000 m³ d⁻¹. Some data points in the piezometers and pumping
wells data show evaporation process. The water in the wells is collected through an access tap connected to the pumping system. For the piezometer water, a pump is used to collect the water and a purge is performed corresponding to 3 piezometer volume. On the measurement chronicle covering a seasonal cycle, the Rhône does not show an evaporative signal. Considering the estimation of the recharge, this evaporative signature is unlikely due to the recharge. This evaporative signature is therefore probably produced by direct evaporation from the aquifer, with an unsaturated zone only 4
to 5 m thick, and in a Mediterranean climatic context. Therefore, this evaporative signature does not allow a rigorous 1D analysis of the isotopic periodic signal. However, an application of Equation (3) to this study site and isotopic data set is provided as an illustrative example in the supplementary information section. It should be noted that this method can be generally applied to alluvial plains especially in case of pumping if a seasonal isotopic signal exists in the river. This method has the advantage of being fast for sampling and analysis, inexpensive and allows assessing a weakly constrained variable in
hydrogeology which is the dispersivity by means of a natural tracer. Artificial tracers cannot always be used due to the proximity to sensitive sites (industrial, urban).

The water supply to the alluvial aquifer by the Ouvèze is then clearly confirmed by radon, isotopic and piezometric data, with the radon activities of the piezometers close to the Ouvèze showing a decrease in activity, as for the Rhône, and a similar isotopic composition between the piezometer RV3 located outside the site (to the east) and the Ouvèze. The use of a
simple 1D solution for mass and heat transport allowed determining the flow velocities via the thermal monitoring. Hence, based on three piezometers the pore velocity was found in the order of $10^{-5}$ m s⁻¹. This value was used for an exploratory interpretation of the groundwater periodic isotopic signal. After calibration using PEST, constrained by the piezometric levels monitoring, the numerical model reproduced the natural flow, i.e., with and without pumping, with velocities consistent with those deduced from temperature signal interpretation. The transport modeling of the radon supports the
feeding of the alluvial aquifer by the Ouvèze, as shown by the intrusion of radon-depleted water sampled at the piezometers near the river, as well as the pumping of the site by the Rhône. Simulated groundwater balance analysis makes it possible to determine a full quantification of water supplied to the site's pumping systems. Therefore, around 38% of the water exploited at the pumping wells comes from the Rhône, 56% from the Ouvèze and 6% from the local recharge associated with rainfall. This groundwater budget is consistent with the isotopic analysis, confirming an important contribution of the
Ouvèze. The simulations confirm a clear vulnerability of the site with respect to possible contaminations from the Rhône and the Ouvèze. The Rhône is extremely close to the pumping wells and remains the main threat regarding potential contamination issues.





## 5 Conclusion

In this study, the ability of a multi-tracer approach to quantify groundwater-river exchanges in a riverside pumping context connected to the aquifer was investigated. The objective was to quantify the river-aquifer interactions at the local scale using a set of tracers in order to obtain robust information to feed a numerical model. This river-aquifer exchange system was simulated using the MODFLOW model under natural and pumping conditions.

Radon has been used in a new way to trace the entry of river water with low radon activity into the aquifer, thus diluting the
natural signal of the aquifer. This rarely used method allows tracing the exchange from the river towards the aquifer, whereas radon is mainly used to determine the groundwater discharge into surface water bodies. This method enables identifying the entry of surface water until a distance from the riverbank of about 50 meters.

The use of the 1D transport model using a periodic input signal allowed the heat flux to be exploited horizontally to obtain an estimate of groundwater pore velocities. The combination of the piezometric, Radon and isotope monitoring showed that
both the Rhône and the Ouvèze rivers recharge the aquifer.

The hydrogeological modeling allowed reproducing the set of observations and to precisely determine the water balance at the pumping site with 38% of water coming from the Rhône and 56% from the Ouvèze. This water balance highlights the vulnerability regarding possible pollutants from the two rivers. It is interesting to note that it may appear somewhat counterintuitive that the smaller river makes greater contributions to the aquifer recharge than the larger river, and indeed the
water management company held that belief before this study. Despite the lower contribution of the Rhône (38%), the greater proximity represents an important risk due to the shorter time to reach the wells and therefore a lower chance of retention or degradation of pollutants.

Beyond our case study, this study confirms the great interest of the radon tracer in the context of river-aquifer exchanges when piezometers close to the riverbank are available, whatever the direction of the exchange. The use of different natural
tracers such as temperature, isotopes or other natural tracers allows for a more accurate estimation of the hydraulic parameter and the creation of a more robust model and is especially interesting where the use of artificial tracers is not an option. Integrated tracer-modeling studies can thus help improve the understanding of aquifer functioning and contribute to a sustainable exploitation strategy.







**Appendices**

As pointed in Section 3.4, the groundwater isotopic signature is marked by evaporation processes preventing a fully reliable interpretation. The application proposed here can therefore only be illustrative and exploratory. The isotopic data for the period without pumping interruption i.e., from October 2018 to January 2020 (Figure 8), are here interpreted using the 1D

solution for a periodic sinusoidal isotopic signal introduced in Section 2. Two piezometers Prg7 and Prg2 were sampled during this period. The river isotopic data show a seasonal signal with a maximum in January and a minimum in June. A fixed pore velocity corresponding to the average value obtained from the interpretation of the temperature signals, i.e., $3.5.10^{-5}$ m s$^{-1}$, constant values for the other parameters (porosity, distances, P) were also used. Consequently, only dispersivity was calibrated, giving a value at 10 m for the isotopic signature of the modeled groundwater (green curve,

Appendix A).

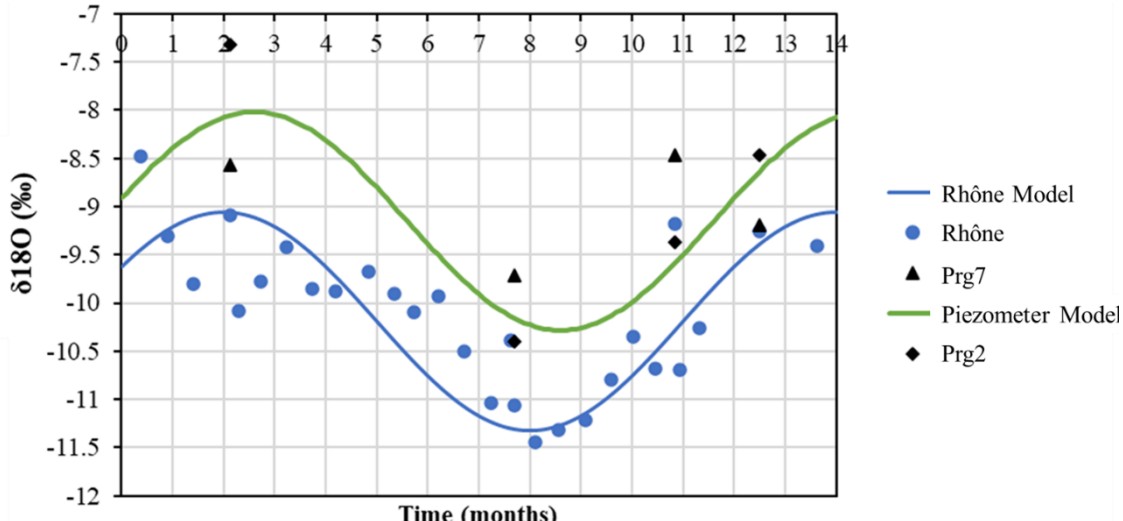

**Appendix A : Stable isotopes of water monitoring at piezometers Prg2 and Prg7 and for the Rhône River and calibrated analytical solutions for a periodic input signal. The Rhône's isotopic data are obtained from Poulain et al. (2021a).**




## Author contribution

J. Texier prepared the manuscript with contributions from all co-authors. J. Texier perform the MODFLOW model with contributions from J. Gonçalves. Field work and data acquisition performed by all co-authors. Isotopic data analysis performed by J. Texier and C. Vallet-Coulomb. Radon data analysis performed by J. Texier and T. Stieglitz.

## Competing interests

The authors declare that they have no conflict of interest.

## Acknowledgments

The Agence de l'Eau Rhône-Meditérannée-Corse, the Syndicat Rhône-Ventoux and SUEZ are acknowledged for their financial support. The modeling software use is Processing MODFLOW X ver.

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
