# Peer review of "Groundwater-Surface water exchanges in an alluvial plain subjected to pumping: a coupled multitracer and modeling approach"

_Hydrology and Earth System Sciences, 2023_

## Referee Comment (RC1)

Review of "*Groundwater-surface water exchanges in an alluvial plain subjected to pumping: a coupled multitracer and modeling approach*" by Texier et al.

The paper is generally well written (although I got lost in a few paragraphs). The main purpose of this paper is to introduce a multi-tracer approach to quantify GW-SW exchange. I fully agree with the authors that complementary information from tracers (and especially natural tracers) can (and should) be used more often in such alluvial river-aquifer contexts to better constrain decision-based model predictions. However, I must admit that I was disappointed to see that the information from the tracers was not really valorized in the modelling exercise. The model was only calibrated against steady-state hydraulic head.

More generally, the paper lacks details on the modeling setup which make it very difficult to understand the main goal of the model (what do they want to predict??). Overall, the so-called approach is not clear to me.

My last concern is that I also felt that the authors did not perform a complete literature review in the introduction. The authors stated that the use of radon in transport models is rarely discussed, but I strongly disagree. The authors are missing some relevant papers. 222Rn (and natural tracers in general) have been used extensively to study river-groundwater interactions under losing river conditions.

Please see the following not exhaustive list of publications: Bertin and Bourg, 1994; Hoehn and Cirpka, 2006; Hoehn and Von Gunten, 1989; Hoehn et al., 1992; Popp et al., 2021; Stellato et al., 2013; Vogt et al., 2010.
See also Peel et al., 2022, Gilfedder et al., 2019, Liao et al., 2021, and Delottier et al., 2022 for explicit simulation of tracers.

In the end, I am not really sure where is the scientific contribution of that paper. In the present form, it is not really clear. For these reasons, I cannot recommend publication of that paper in HESS.

Detailed comments

**Line 28:** PEST suite.

**Line 29:** Is that really reactive transport for Radon ?

Groundwater-river; aquifer-surface water etc. Please be consistent in the paper.

**Line 108:** I would just say a calibrated model. If a model is badly calibrated, it is better to say that it is not calibrated.

**Line 114**: tracers are observations (not techniques). Here the author refer to method and technique but I think it is observations right ?

**Figure 1:** Not really easy to see where are pumping wells and where are piezometers. Need more detailed legend.

After reading Part 2.1, I am still not sure about the location of the pumping wells. For the aquifer geometry, a geological cross section would be welcome.

**Line 226:** Specific yield

**Line 220:** Why do you name it a synthetic model? Is that not a model developed in a real case study?

**Line 236:** Is the Rhone river represented with a Dirichlet BC ? If so, this can lead to enormous amount of water in the model. Again, the description of the model is not so clear. Why not used a Cauchy type BC?

**Lines 238,239,240:** This means that there is only one layer for the entire model? So this is a pseudo 2D model?

**Line 240:** permeability field? Is this considered homogeneous or Heterogeneous? If so, is there zones of piecewise constancy or pilot points? Not clear. How many parameters involved in the model calibration?

**Line 242:** PEST optimization tool. This means that you have used the CMAES global optimisation scheme? Not clear.

**Line 245:** Not production of radon in the groundwater? Not clear. How the production of radon can be simulated with an injection well? More information is needed here to better understand how Radon was simulated in the model.

**Line 250:** This is not a classical way to simulate radon. Usually an end-member equilibrium activity is needed for that seeks.

**Figure 3:** The use of an inverse distance method to draw a piezometric map is not ideal. The piezometric map seams strange with geometric 90° change of orientation. Is this because of the method or because of strong heterogeneity at the site scale?

**Figure 4:** Not clear. What is the meaning of the grey circle in the Figure? The large grey band ? Not easy to follow.

**Line 297:** explain the meaning of GMWL (global meteoric water line I guess).

**Line 303:** meteoric groundwater recharge.

**Line 335:** this method? I should admit that I am getting lost in the end of this paragraph.

**Section 3.4:** Is there any uncertainty on the temperature models used to interpret the data? It would be good to discuss the reliability of the results of these models regarding the uncertainty.

**Figure 8:** I don't find the C) section in this figure. This is unfortunate since I also find that the A) and B) are far from being informative to support model calibration.

**Line 350:** I do not see how the calibrated model reproduces the observed data. This is not clear at all in the figure.

**Section 3.5.1:** How the results of the model are sensitive to the estimated parameters? At least, a sensitivity analysis should be applied.

**Section 3.5.2**: As I understand, the radon and isotopic information were not added at all in the model calibration? Why?

References

Bertin, C., Bourg, A.C., 1994. Radon-222 and chloride as natural tracers of the infiltration of river water into an alluvial aquifer in which there is significant river/groundwater mixing. Environ. Sci. Technol. 28 (5), 794–798. https://doi.org/10.1021/ es00054a008.

Hoehn, E., Cirpka, O.A., 2006. Assessing residence times of hyporheic ground water in two alluvial flood plains of the Southern Alps using water temperature and tracers. Hydrol. Earth Syst. Sci. 10 (4), 553–563. https://doi.org/10.5194/hess-10-553-2006.

Hoehn, E., Von Gunten, H.R., 1989. Radon in groundwater: a tool to assess infiltration from surface waters to aquifers. Water Resour. Res. 25 (8), 1795–1803. https://doi.org/10.1029/WR025i008p01795.

Hoehn, E., Von Gunten, H.R., Stauffer, F., Dracos, T., 1992. Radon-222 as a groundwater tracer. A laboratory study. Environ. Sci. Technol. 26 (4), 734–738. https://doi.org/ 10.1021/es00028a010.

Popp, A.L., et al., 2021. A Framework for Untangling Transient Groundwater Mixing and Travel Times. Water Resour. Res. 57 (4) https://doi.org/10.1029/2020wr028362.

Stellato, L., et al., 2013. Is 222Rn a suitable tracer of stream–groundwater interactions? A case study in Central Italy. Appl. Geochem. 32, 108–117. https://doi.org/10.1016/j. apgeochem.2012.08.022.

Vogt, T., et al., 2010. Fluctuations of electrical conductivity as a natural tracer for bank filtration in a losing stream. Adv. Water Resour. 33 (11), 1296–1308. https://doi.org/10.1016/j.advwatres.2010.02.007.

Gilfedder, B., Cartwright, I., Hofmann, H., and Frei, S. (2019). Explicit modeling of Radon-222 in HydroGeoSphere during steady state and dynamic transient storage. Groundwater 57, 36–47. doi: 10.1111/gwat.12847

Delottier, H., Peel, M., Musy, S., Schilling, O.S., Purtschert, R., Brunner, P., 2022. Explicit simulation of environmental gas tracers with integrated surface and subsurface hydrological models. Front. Water 4, 1–12.

Liao, F., Cardenas, M.B., Ferencz, S.B., Chen, X., Wang, G., 2021. Tracing bank storage and hyporheic exchange dynamics using 222Rn: virtual and field tests and comparison with other tracers. Water Resour. Res. 57 (5), e2020WR028960.

Peel, M., Kipfer, R., Hunkeler, D., Brunner, P., 2022. Variable 222Rn emanation rates in an alluvial aquifer: limits on using 222Rn as a tracer of surface water–Groundwater interactions. Chem. Geol. 120829.

---

## Author Comment (AC1)

**RC1**

GC1) The paper is generally well written (although I got lost in a few paragraphs). The main purpose of this paper is to introduce a multi-tracer approach to quantify GW-SW exchange. I fully agree with the authors that complementary information from tracers (and especially natural tracers) can (and should) be used more often in such alluvial river-aquifer contexts to better constrain decision-based model predictions.

Thank you for your thoughtful and constructive review on our paper. We appreciate your positive assessment of the overall writing quality and your acknowledgment of the importance of a multi-tracer approach in quantifying groundwater-surface water (GW-SW) exchange in alluvial river-aquifer systems. We understand your concerns regarding the clarity of certain paragraphs and proceeded to a major revision of those unclear sections to enhance overall readability.

GC2) However, I must admit that I was disappointed to see that the information from the tracers was not really valorized in the modelling exercise. The model was only calibrated against steady state hydraulic head.

The model has been effectively calibrated on hydraulic heads; we have not calibrated any transport parameters. We just added a production parameter to reproduce the natural value (without pumping) of radon in groundwater. With pumping, we just verified that the model was able to reproduce radon observations without any additional calibration (in a parsimonious approach). This validates the global flow and transport model. Regarding stable isotopes of water, they allow identifying a mixing (surface water-groundwater exchanges) which is a major contribution to the conceptualization of the system in support of a numerical model (in terms of boundary conditions).

From above statement and the response to comment GC3 below, we believe that we have exploited the tracers in a relatively efficient way.

GC3) More generally, the paper lacks details on the modeling setup which make it very difficult to understand the main goal of the model (what do they want to predict??). Overall, the so-called approach is not clear to me.

To address this, we now provide more details on our modeling approach in the manuscript (some information also added as responses to the detailed comment's part, see below). Additionally, we clarified our goal and workflow approach in the introduction. The main goal of this study is to determine the origin and proportions of the water sources feeding a pumping well system on an alluvial plain. For this purpose, the following steps were implemented:

i) Identification of the water sources feeding the pumping wells using at least two distinct tracers (radon and stable water isotopes) and the piezometric data.

ii) Analysis of temperature seasonality in the piezometers and Rhône River to estimate pore velocity (u), based on known porosity.

iii) Use of $\delta 18O$ seasonality in piezometers to constrain dispersivity, a crucial but often poorly known property.

iv) Use of steady-state flow modelling coupled with reactive transport (radon) to confirm the origins and quantify the proportions of the pumped water mixture, using pore velocity and radon spatial distribution only for model validation.

During the study the δ18O seasonality analysis in piezometers presented in the additional content section, was not as robust as expected and the dispersivity value obtained can only considered as a first order estimate. However, this value in the order of 10 m is in excellent agreement with those reported in the literature for similar media (Schulze-Makuch, 2005).

GC4) My last concern is that I also felt that the authors did not perform a complete literature review in the introduction. The authors stated that the use of radon in transport models is rarely discussed, but I strongly disagree. The authors are missing some relevant papers. 222Rn (and natural tracers in general) have been used extensively to study river-groundwater interactions under losing river conditions. Please see the following not exhaustive list of publications: Bertin and Bourg, 1994; Hoehn and Cirpka, 2006; Hoehn and Von Gunten, 1989; Hoehn et al., 1992; Popp et al., 2021; Stellato et al., 2013; Vogt et al., 2010. See also Peel et al., 2022, Gilfedder et al., 2019, Liao et al., 2021, and Delottier et al., 2022 for explicit simulation of tracers.

In the previous version of the manuscript, the novelty approach regarding radon was introduced once with a certain moderation (line 87-90 and see below) and secondly in an inadequate manner (line 133-137), the latter being comprehensibly the only one commented on by RC1. In fact, the moderated paragraph in the previous manuscript was supported by some literature, one of which is mentioned by RC1 (Hoehn and Von Gunten, 1989) and one of which is not (Close et al., 2014). In the revised version we retained the moderated presentation (using "less studied" for losing-river radon use and radon modeling) adding some of the references pointed by RC1, for which we are grateful.

Among omitted literature, the work by Adyasari et al., (2023) allows discussing the real importance of radon used under losing river conditions. Similarly to these authors, alternative requests using the keywords "groundwater discharge", "tracer", "radon" (gaining river); and "river infiltration", "radon" (losing river) performed on Google Scholar and Web of Science led to about 15-20% of the paper dedicated to losing river situations (on a total of between 100 and 130 articles between 2003 and 2023). Based on this analysis, gaining river situation are largely dominating and, in our opinion, the term "extensively" doesn't really apply to the use of radon to losing river situation. This is exactly what is said in the moderated version: "However, the situation of a losing river, i.e., surface water supplying the groundwater is less studied."(Line 87 of the previous discussion version of the manuscript). This is likely related to the fact that radon data interpretation using a standard and simple mixing reactive model approach for surface waters is obviously adapted to gaining rivers (or lakes). More complex calculations (groundwater flow and reactive transport modeling) are required for losing rivers. In the same way, we stated lines 136-137 (previous manuscript)," the use of radon data in transport models is rarely discussed"

In addition to the abovementioned moderation now adopted, the uncommon situation of interest here which can be considered as a novelty was added in the introduction. Previous studies typically focus solely on either "gaining river" or "losing river" situations. However, in this study, alternating gaining and losing river situations occur at the same site allowing the identification of groundwater sources and proportions.

GC5) In the end, I am not really sure where is the scientific contribution of that paper. In the present form, it is not really clear. For these reasons, I cannot recommend publication of that paper in HESS.

We hope that we have answered the major concerns of Rev1 (see responses to GC1 to 5). Overall, we have considered all the constructive comments made by reviewers, which has resulted in a greatly improved manuscript. Hopefully, the revised manuscript will better meet your expectations of HESS.

We believe that the scientific contribution was clarified in the respond to comment GC3 with the goal (subject of public interest due to the common situation considered) and the methodology (workflow) involving the sequential use of tracers to obtain parameters (dispersivity with $\delta^{18}O$), variables (pore velocity with temperature), the conceptual model ($\delta^{18}O$ and radon) to obtain a more robust numerical model enhancing the confidence on model outputs in terms of groundwater balance (mixing proportions in wells). The goal and the workflow are repeatable in this common situation of pumping facilities in alluvial plain system.

Detailed comments

1) Line 28: PEST suite.

Corrected accordingly.

2) Line 29: Is that really reactive transport for Radon ?

Yes, we consider a reactive transport which is the case when a chemical element is transformed or degraded during its transport. In our case, the radon is generated in the aquifer and is degraded by radioactive decay.

3) Groundwater-river; aquifer-surface water etc. Please be consistent in the paper.

Corrected accordingly with groundwater-surface water.

4) Line 108: I would just say a calibrated model. If a model is badly calibrated, it is better to say that it is not calibrated.

Corrected accordingly.

5) Line 114: tracers are observations (not techniques). Here the author refer to method and technique but I think it is observations right ?

Yes, indeed we made measurements that are interpreted using models. This was corrected in the manuscript accordingly.

6) Figure 1: Not really easy to see where are pumping wells and where are piezometers. Need more detailed legend. After reading Part 2.1, I am still not sure about the location of the pumping wells. For the aquifer geometry, a geological cross section would be welcome.

Figure 1 was corrected with a legend for the different symbols used and better style for the surface water. A cross section was added into figure 3 to explain the mechanism of exchanges.

7) Line 226: Specific yield

Corrected accordingly.

8) Line 220: Why do you name it a synthetic model? Is that not a model developed in a real case study?

Corrected using "model". This model is indeed developed using a real case study.

9) Line 236: Is the Rhone river represented with a Dirichlet BC ? If so, this can lead to enormous amount of water in the model. Again, the description of the model is not so clear. Why not used a Cauchy type BC?

The river stages are roughly stable which can be conveniently described using Dirichlet BC. With such conditions, there is no need to calibrate a (generally poorly constrained) conductance coefficient. The "enormous amount" of water behind the Dirichlet BC simply correspond to the supply the pumping wells since the overall mass balance is met (by definition) in the model.

10) Lines 238,239,240: This means that there is only one layer for the entire model? So this is a pseudo 2D model?

. The hydraulic head and the radon activity simulated here are functions of space coordinates x and y but not z (h(x,y), R(x,y)). It is therefore a pseudo 3D model but actually a real 2D model.

11) Line 240: permeability field? Is this considered homogeneous or Heterogeneous? If so, is there zones of piecewise constancy or pilot points? Not clear. How many parameters involved in the model calibration?

The permeability field being heterogeneous, the domain is separated in several zones of piecewise constancy. During the calibration 10 zones were used for 10 observation points. The information was added line 253-256.

12) Line 242: PEST optimization tool. This means that you have used the CMAES global optimisation scheme? Not clear.

We use the module PEST incorporated in our MODFLOW version (processing MODFLOW X). PEST uses a nonlinear estimation technique known as the Gauss-Marquardt-Evenberg method which is a standard gradient-based optimization algorithm.

13) Line 245: Not production of radon in the groundwater? Not clear. How the production of radon can be simulated with an injection well? More information is needed here to better understand how Radon was simulated in the model.

Radon geological production was added using the "injection well" package of Modflow. Injection wells are implemented in all aquifer cells, with a very low injection rate to avoid artificial impact on the water table, and a large mass of chemicals to reproduce radon generation. This injected mass is considered as homogeneous, and its value is set to reproduce natural radon activity. Radon radioactive decay is also implemented. Details on this technical implementation of the production are provided lines 260-264.

14) Line 250: This is not a classical way to simulate radon. Usually an end-member equilibrium activity is needed for that seeks.

Unfortunately end member equilibrium is not available on our version of MODFLOW and MT3D.

15) Figure 3: The use of an inverse distance method to draw a piezometric map is not ideal. The piezometric map seams strange with geometric 90° change of orientation. Is this because of the method or because of strong heterogeneity at the site scale?

The appearance is due to the interpolation method and the software. We change it and the figure 3 was redrawn accordingly. Additionally, we added the piezometric map without pumping.

16) Figure 4: Not clear. What is the meaning of the grey circle in the Figure? The large grey band ? Not easy to follow.

Figure 4 and the description was modified accordingly to a better understanding. The large grey band corresponds to a rupture in the x axis. The grey circle square in this (new manuscript) correspond to piezometer Prg2 during the pumping stopped period.

17) Line 297: explain the meaning of GMWL (global meteoric water line I guess).

GMWL is indeed global meteoric water line the definition was added line 314.

18) Line 303: meteoric groundwater recharge.

Corrected accordingly.

19) Line 335: this method? I should admit that I am getting lost in the end of this paragraph. Section 3.4: Is there any uncertainty on the temperature models used to interpret the data? It would be good to discuss the reliability of the results of these models regarding the uncertainty.

A sensitivity analysis was added in this section to determine the relative importance of the parameters. It appears that he flows velocity is the more sensitive one (line 351, figure 7A). Additionally, we also present the limitation of this model and alternative approaches when an analytical solution is not applicable (lines 205-209).

20) Figure 8: I don't find the C) section in this figure. This is unfortunate since I also find that the A) and B) are far from being informative to support model calibration.

Corrected accordingly, figure C is restored.

21) Line 350: I do not see how the calibrated model reproduces the observed data. This is not clear at all in the figure. Section 3.5.1: How the results of the model are sensitive to the estimated parameters? At least, a sensitivity analysis should be applied. Section 3.5.2: As I understand, the radon and isotopic information were not added at all in the model calibration? Why?

The calibration result is added in the scattering plot in figure C that was absent in the original version of the manuscript. Additionally, the calibration is effectively only made using hydraulic head data. However, we reproduce the observed radon activity as explained lines

388-390 and figure 9, indicating a good estimation of the flow parameters as explained with more details in our response to GC3.

References

Adyasari, D., Dimova, N. T., Dulai, H., Gilfedder, B. S., Cartwright, I., McKenzie, T., and Fuleky, P.: Radon-222 as a groundwater discharge tracer to surface waters, Earth-Science Rev., 238, 104321, https://doi.org/10.1016/j.earscirev.2023.104321, 2023.

Schulze-Makuch, D.: Longitudinal dispersivity data and implications for scaling behavior, Ground Water, 43, 443–456, https://doi.org/10.1111/j.1745-6584.2005.0051.x, 2005.

---

## Author Comment (AC2)

**RC2**

GC1) I reviewer the paper titled "Groundwater-Surface water exchanges in an alluvial plain subjected to pumping: a coupled multitracer and modeling approach". This study proposes a mulitracer approach as well as modelling to understand the stream/groundwater interactions in an alluvial plain. The particularity of the study site is that pumping wells are located near the stream and the water extraction controls the stream/groundwater interactions and the groundwater flow in the alluvial aquifer. The subject of the study is interesting as the groundwater extraction in alluvial aquifers near streams can have major impacts on the stream ecosystem. An other interesting point is that the groundwater extraction is located between two streams, which both contribute to the aquifer recharge imposed through pumping.

Thank you for your detailed and thoughtful review of the paper. We appreciate the time and effort you invested in evaluating our work. Your constructive comments provide valuable insights that undoubtedly contribute to refining and enhancing the scientific merit of our study.

GC2) We understand that authors made efforts to collect field data and to use different approaches and methods, which is always worthy. However, I have major issues with the study and the data interpretation. First of all, I really wonder what is the novelty of the study. The methods used in the study have been widely applied in the context of stream/groundwater interactions and the scientific contribution is not clear at all. Then, it can be a good point to use different methods. However, in this case, it does not seem relevant. The fact that the pumping controls the groundwater flow and the stream/groundwater interactions is obvious and can be seen only using piezometric levels.

The novelty issues are discussed in response to GC4 of Rev1 (not reproduced here, if you don't mind referring back to it) and GC3 below.

Regarding piezometric levels, it is true that they usually give valuable information and allow determining the flow direction. However, streamline coming from a river does not necessarily mean a losing river (and therefore a connection) if the river is completely clogged. The clogging situation, which is not the common situation, was considered in a previous hydrogeological modeling of the study area not discussed here. The practitioners we were in contact with were wondering about this conceptualization of a river disconnected from the alluvial plain, which has a direct impact on the management of the pumping field in terms of water quality. This maybe brings some useful clarification to justify the (from zero) approach followed here.

In this context, the use of tracers allows to identify surface water contribution and to update the conceptual model of groundwater-surface exchange. Additionally, to obtain other information as flow velocity or dispersivity, to quantity the different water sources of pumping wells, other information is needed. The detail of the workflow is proposed just below.

GC3) The others methods used (isotope, radon…) lead to the same conclusion and it is not clear how the different methods are complementary. Likewise, the model could be interesting but its potential is not fully used. We have the feeling that the Modflow model only allows reproducing the piezometer levels. It should have been more interesting to discuss the integration of the data collected in a model.

To address this, we detail our modeling approach in the manuscript (the details were added as responses to the detailed comment's below). Additionally, we clarified our goal and workflow in the introduction. The main goal of this study is to determine the origin and proportions of the various water sources feeding a pumping well system in an alluvial plain. For this purpose, the following steps were implemented:

i) Identification of the water sources feeding the pumping wells using at least two distinct tracers (radon and stable water isotopes) and the piezometric data.

ii) Analysis of temperature seasonality in the piezometers and Rhône River to estimate pore velocity (u), based on known porosity.

iii) Use of $\delta 18O$ seasonality in piezometers to constrain dispersivity, a crucial but often poorly known property.

iv) Use of steady-state flow modelling coupled with reactive transport (radon) to confirm the origins and quantify the proportions of the pumped water mixture, using pore velocity and radon spatial distribution only for model validation.

During the study the $\delta 18O$ seasonality analysis in piezometers presented in the additional content section, was not as robust as expected and the dispersivity value obtained can only considered as a first order estimate. However, this value in the order of 10 m is in excellent agreement with those reported in the literature for similar media (Schulze-Makuch, 2005).

GC4) In addition, I found that the text is not always clear (and is confusing at some points) and that Figures could be improved for a better understanding. I also have major concerns about the data interpretation of the thermal signals (see details below) and I am not sure that the results are consistent and relevant.

Detailed responses are provided in the detailed section below.

**GC5) To summarize, I would say that the manuscript could be considerably improved. The methods and the data interpretation could go further. Most importantly, the complementarity of the methods should be strengthen. In this present form, the manuscript seems to be a succession of data that are not fully used and enhanced. I am not sure that the scientific contribution is enough for a publication in HESS and I would recommend to submit it in a journal specialized in regional studies.**

Overall, we have considered all the constructive comments made by reviewers, which has resulted in a greatly improved manuscript. Hopefully, the revised manuscript will better meet your expectations of HESS.

**Specific comments**

In addition to the major issues mentioned above, here are some specific comments that could help improve the manuscript :

1) The introduction should be significantly improved. It is not pleasant to read as it is a list of methods that can be used. It would be interesting to better explain the purpose of the use of each method. The last paragraph is way too long and not clear at all. It

seems there are some results in it and at the end, we don't understand the goal of the study.

- o The introduction was partially modified. We still present the different methods used and their interplay (workflow). But we understand your concern about the purpose of our study, and additional information on the goal and workflow was added for better clarity (see also response to GC3).

2) I recommend to add a section that clearly explains the goal of each approach, how the data collected will be used and why you have chosen these approaches.

- o The detailed goal and workflow were added in the introduction part line118-134.

3) The section 2.2 is not clear at all.

- o The section 2.2 was rewritten accordingly.

4) 200 the solutions of Goto and Stallman have been adapted … what does this imply ? Are there references ? I am sure that you can find in the literature some authors who used the temperature signal in a purely horizontal case.

- o The Goto and Stalman's solutions were originally used for vertical heat transport. Here, it is used   in the context of a horizontal groundwater flow. The works are references:  Goto and Stalman are cited 174 times and 620 times respectively. We cited in the introduction some articles interpreting temperatures by modeling in horizontal case (71-75) and we added other references. Additionally, we modify these lines for better clarity.

5) Section 2.3.1 / section 2.3.2 – what is the purpose of these methods ? what parameters will be estimated from these solutions ? At what spatial resolution ?

- o As explained in lines 209 and 225 these solutions were used to obtain the pore velocity and the dispersion coefficient (linked to dispersivity) by inversion.

6) 205 . I don't understand this sentence. Can you explain how thermal properties are defined exactly ?

- o Thermal properties which refer in fact to the effective thermal diffusivity are obtained by numerical inversion using the temperature records.

7) Figure 2 . We don't clearly see the data of Prg7 and Prg1. Please, change the color of Prg1 or Rhône (using blue for both is not practical). In the aquifer, the flow direction don't vary in time ? The GW level is always : Prg2>Prg1>Prg6>Prg7 ? Also, can you add the stream level in the Ouvèze ?

- o Figure 2 was modified accordingly and the variation of the Ouvèze stage was added. Yes, we explain that during the activity of the site the hydrodynamics are weakly variable. This is the reason why so we considered a pseudo steady state.

8) Figure 3 should be better presented and discussed (only one sentence about it). In the present form, the figure seems useless. It would be interesting to show one map under pumping and one map without pumping.

- o Figure 3 was modified to present the pumping case and the natural case, with schematic view of the aquifer- river interactions. Figure 3 is now presented with figure 2 to allow a better view of the impact of the piezometric variations.

9) One interesting point to discuss and interpret is the change observed when the pumping stops.

- o The change of hydraulic conditions is now more deeply discussed in Section 3.1 and the new version of Figure 3.

10) Figures 4 and 5 are incomprehensible… you speak about three campaigns but we don't see these results in the Figure (different colors for the different campaigns should be good); The legends do not correspond with the data presented in the Figure (shape and colors); It should be nice to add the name of piezometers to better visualize the position of the measurements; how was estimated the distance from the Rhone ? what is the reference ? From Figure 1 and 3, I don't see how you can have data between 650 and 750 m ? ; the points at streams seem to be 0. Did you try a log-scale ? Likewise, the text from L. 275 should be revised because, in the present form, the text does not reflect what we see in Figure 4 – for instance, I don't see the different campaigns…

- o We updated Figures 3 and 4 as recommended by Rev1 and Rev2. The piezometer name was added on the points to allow a better understanding. Distances are estimated perpendicular to the Rhone river with the origin at the river bank (i.e. Rhone = 0m). The distances were obtained using a GIS software and the coordinate of the piezometer and wells. The different campaigns are now represented on the two figures using different symbols (square campaign 1, triangle campaign 2, circle campaign 3). The colors in Figure 4 represent the state of the pumping site (green=pumping, grey=pumping stopped). In this configuration the sampling site at the Ouvèze is located 750 m from the Rhone (as seen from Figure 1b).

11) Section 3.3 – do you have the same results with and without pumping ?

- o No isotopic data available for this period.

12) For the interpretation of temperature data, there are several critical points.

- o You have one equation and several unknown parameters – how did you manage that ?

  - ▪ There is one equation but a times-series of temperature measurements to calibrate the two parameters (the same occurs when calibrating storage coefficient and transmissivity using time series of drawdowns and e.g. the Theis solution for pumping tests).

- o The value of the thermal conductivity is a critical point when you interpret thermal curve (see for instance 10.5194/hess-26-1459-2022). How did you whose the value of the thermal conductivity (which is included in the thermal diffusivity) ? it would have be nice to have a sensitivity analysis to see the flux interpretation for different values of thermal diffusivity.

  - ▪ Although Rev2 is right regarding the importance of thermal conductivity, in fact only the effective thermal diffusivity (ratio of bulk thermal conductivity to bulk thermal capacity of the porous media) was calibrated. As suggested, we performed a sensitivity analysis for the two calibrated parameters (the pore velocity and the effective thermal diffusivity). The analysis shows that the pore velocity is the more sensitive parameter (lines 351-353, Figure 7B).

o In understand that under pumping conditions, the hydraulic gradient is quasi-constant. However, during the pumping stop, the GW flux and the flow direction changed. In this case, how is it possible, with only one value of flux, to model the entire period? This result does not seem consistent to me.

o Only steady state calculations corresponding alternatively to natural or pumping situation were performed. The model is calibrated in steady state by using the pumping period and the pore velocity obtained by temperature seasonality interpretation is only used for model validation in this situation (pumping).

o The estimated values of thermal diffusivity don't seem consistent to me. In the literature, we can find in Stauffer 2013 that values of thermal diffusivity vary between 3.2x10-7 and 1.8e-6 (for all materials – clay, silt, sand, gravel, blocks) (for instance : Stauffer, F., Bayer, P., Blum, P., Molina Giraldo, N., &Kinzelbach, W. (2013). Thermal Use of Shallow Groundwater. https://doi.org/10.1201/b16239), the values of thermal diffusivity very) . This may explain that the value of flux are not consistent with Darcy results.

  ▪ Rev2 is perfectly right, and we are grateful for this suggestion and the reference. Diffusivity values were indeed incorrect. We proceeded to a more careful parameters space exploration by using reference values as boundaries to avoid selection of local minimum of the objective function (here RMSE). This was also the time for a sensitivity analysis besides the identification of realistic optimal parameter values. Additionally, we corrected a mistake in the darcy flow velocity calculation with a value now in between $10^{-4}$ and $10^{-5}$ m s$^{-1}$. Therefore, the velocity obtained using the thermal signal analysis are slightly lower but more consistent with the calculated values using MODFLOW.

o At last, I would recommend to interpret the temperature signal with other approachs and tools – for instance using FLUX-BOT http://dx.doi.org/10.1002/hyp.11198 - to compare your results

  ▪ The comparison was not made using FLUX-BOT, but we added in the reference at line 207 a reference to the use of numerical model for more complex case study than ours. Additionally, the adaptation of the analytical solution (by change of the direction z to x) does not change its validity.

o The authors should better highlight the limits of the model/approach (there are many unknown parameters…)

  ▪ The limit of the approach and the conditions were commented line 203-208 and we demonstrate with the sensitivity analysis that the pore fluid velocity is the most sensitive and that we found the optimal value in the space parameter.

13) The authors should better explained the interest of the model… Section 3.5.1 What is the interest of the model ? Apart from reproducing the piezometric maps. Fig 9 : what is the interest of the radon transport model ? What are the new results (compared to piezometric levels) ?

- o The model allows us to obtain a relevant estimate of the unknown water proportion supplying the pumping wells. We considered this information important in the context of contamination vulnerability regarding the pumping installation. (More detail in response to GC 2 and GC3 concerning the interest of our approaches)

Some minors comments/suggestions :

- 51 : The issue with the Darcy method is also that it is integrative in space.
  - o This precision was added accordingly.
- 54 "several orders of magnitude" – in space or time ?
  - o Refers to differences in space.
- 56 what are the advantages and disadvantages of this approach ?
  - o Some precision was added. Line 54-58
- 60 Artifical tracers should be defined here
  - o Sentences modified to add a short definition.
- 104 . Can you rephrase this ? It is not clear
  - o Rephrased accordingly.
- 125 – defined 'thermal parameters of the environment'
  - o Thermal parameters refer here to the effective thermal diffusivity of the medium, corrected accordingly.
- Fig 1 . The name of the streams are not visible. A legend should be used to say what are yellow triangles and red points. I know this is written in the caption but it is not practical for the reader.
  - o Figure 1 corrected accordingly.
- 149 . What do you mean about "extraction site" ? is it relevant ?
  - o The pumping site corrected accordingly.
- 149 Please make two sentences. What is the flow rate of each stream ?
  - o Corrected accordingly and the flow rates of each stream were added. (line 155-157)
- 158-166 . this is not clear at all. From L;164 – Do you speak about the same piezometer as above ?
  - o Yes, these are the same piezometers, the paragraph was rewritten to be more clear.
- 172 . One campaign means one sample for each point ?
  - o The radon campaign has not the same goal as explained line 172-180. "The first campaign focused on the Rhône riverbank and the pumping site during a period of pumping shutdown. The second occurred when pumping activity resumed and focused on the groundwater abstraction site. The third campaign focused on the Ouvèze riverbank and the aquifer outside the site in the same hydrodynamic context of the second campaign (pumping wells functioning and related pseudo-steady state)."

- 176. Frequency of measurements ?
  - During the period, there were 6 campaigns.
- 191 please specify the unit of the thermal diffusivity
  - Unity added.
- Any reference for Equation 4 ? If not, how did you validate the use of this solution ?
- Equation is an adaptation of the validated analytical expression (Eq. 2) for the temperature noting that heat transport (heat balance) and advection-dispersion equations have exactly the same form, allowing parameters identification.
- 219. The sentence should be rephrased. The purpose of the model is not to integrate the results obtained with the tracers
  - Sentences rewritten accordingly.
- 233 . THE volumetric..
  - Corrected accordingly.
- 237; Can you justify the northern boundary conditions (17 m?)
  - The northern BC is set at 17 m by analysis of previous piezometric maps established at different times indicating a zone around 17 m due to the influence of the Rhône and Ouvèze rivers on the alluvial groundwater table. Unfortunately, there is no geological boundary in the vicinity that could be used as an alternative BC. Justification added line 248-250.
- 244 – what is the purpose of integrating radon transport in the model?
  - The reproduction of the radon transport without further calibration indicates that the hydraulic condition is well reproduced during pumping, confirm a good identification of the hydraulic parameters, and reinforce the confidence in the overall calculated groundwater balance (source and contribution to pumping) which was in fact the main goal of the modeling.
- 254 – what do you mean by "short time scale" ? idem L. 257 for long time
  - Sentences rewritten to be clearer. Short time scale is used for the daily variation of the groundwater.
- 258 – flooding is not a long time scale for me.
  - Indeed, text corrected accordingly.
- 263 – What about the direction of flow relative to the Ouvèze ?
  - Even during the flooding period, the Rhône stage and the groundwater level remain below the Ouvèze leading to a constant direction of the flow from the Ouvèze to the pumping site.
- 263 – Can you clarify what are the "periods of flooding" – it is difficult to see them in the Figure and to estimate their duration.
  - The flooding period refer to the increase of the Rhone level observed on figure 2. These flooding periods vary considerably over time but lead to a Rhône level of 16.8 m (except for the period in November and December 2019, with three major peaks in the Rhône level). The stability of the Rhône level in this area is due to the presence of a dam upstream.

- 274 . The title should be changed – there is no interpretation in this section.
  - Corrected accordingly.
- 276 – where is Prg2 on the Figure ?
  - The piezometer ID was added on the figure 4 and 5.
- 280 – SRV3 id RV3 ?
  - We confirm SRV3 is RV3. typo corrected.
- 295 – please refer to Figure 6
  - Corrected accordingly.
- 298 – what is GMWL ?
  - We corrected accordingly by defining GMWL in the manuscript (GMWL= global meteoric water line)
- Figure – there are several points for each piezometer – what is the frequency of measurements?
  - Not really regular frequence but 5 campaign between June 2019 and November 2020
- 322-323. I don't understand what you mean here.
  - Sentences rewritten accordingly.
- 328 . Is the value of porosity consistent with literature ?
  - We don't find precise information in the literature for this precise location. We used several reports made by or for the operator. Additionally, these values are consistent for the lithology of the area (Nofal, 2014).
- 331 – 333 – where does the values of hydraulic conductivity come from ?
  - The hydraulic conductivity come from pumping test made by the operator of the pumping site and the literature (reference of internal document added).
- 340 – this paragraph should in the discussion section
  - Paragraph moved and integrated to the discussion section.
- 418. 56% is important compared to the size/rate of the stream. Does the infiltration significantly affect the flow rate of the stream ? What is the loss in height ?
  - The pumping site extracts 11,000 $m^3$ $day^{-1}$ (0.12 $m^3$ $s^{-1}$) supplied by the Rhône (400 $m^3$ $s^{-1}$) and the Ouvèze (19 $m^3$ $s^{-1}$). 60% of $m^3$ $s^{-1}$ corresponds to 0.072 $m^3$ $s^{-1}$, in other world 0.4% of the average flow of the Ouvèze. In this case, we don't consider that the infiltration of the Ouvèze into the aquifer is likely to have any visible impact on the river.